# Effectiveness of Extended or Continuous vs. Bolus Infusion of Broad-Spectrum Beta-Lactam Antibiotics for Febrile Neutropenia: A Systematic Review and Meta-Analysis

**DOI:** 10.3390/antibiotics12061024

**Published:** 2023-06-07

**Authors:** Kazuhiro Ishikawa, Koko Shibutani, Fujimi Kawai, Erika Ota, Osamu Takahashi, Nobuyoshi Mori

**Affiliations:** 1Department of Infectious Diseases, St. Luke’s International Hospital, Tokyo 104-8560, Japan; kokoshib@luke.ac.jp (K.S.);; 2Library, Center for Academic Resources, St. Luke’s International University, Tokyo 104-0044, Japan; 3Global Health Nursing, Graduate School of Nursing Sciences, St. Luke’s International University, Tokyo 104-0044, Japan; otae@luke.ac.jp; 4Tokyo Foundation for Policy Research, Tokyo 106-0032, Japan; 5Graduate School of Public Health, St. Luke’s International University, Tokyo 104-0045, Japan; otakahas@luke.ac.jp

**Keywords:** bolus infusion, continuous infusion, extended infusion, febrile neutropenia

## Abstract

This systematic review aimed to compare extended infusion or continuous infusion with bolus infusion for febrile neutropenia (FN). We included clinical trials comparing extended or continuous infusion with bolus infusion of beta-lactam antibiotics as empirical treatment for FN and evaluated the clinical failure, all-cause mortality, and adverse event rates. Five articles (three randomized controlled trials (RCTs) and two retrospective studies) from 2014 to 2022 were included. Clinical failure was assessed with a risk ratio (RR) (95% coincident interval (CI)) of 0.74 (0.53, 1.05) and odds ratio (OR) (95% CI) of 0.14 (0.02, 1.17) in the 2 RCTs and retrospective studies, respectively. All-cause mortality was assessed with an RR (95% CI) of 1.25 (0.44, 3.54) and OR (95% CI) of 1.00 (0.44, 2.23) in the RCTs and retrospective studies, respectively. Only 1 RCT evaluated adverse events (with an RR (95% CI) of 0.46 (0.13, 1.65)). The quality of evidence was “low” for clinical failure and all-cause mortality in the RCTs. In the retrospective studies, the clinical failure and all-cause mortality evidence qualities were considered “very low” due to the study design. Extended or continuous infusion of beta-lactam antibiotics did not reduce mortality better than bolus infusion but was associated with shorter fever durations and fewer adverse events.

## 1. Introduction

Chemotherapy-induced neutropenia is a common problem in cancer patients, with fever being a frequent complication. It affects approximately 10–50% of patients with solid tumors and more than 80% of those with hematologic malignancies [1]. In a retrospective study [2], it was established that the 30-day mortality rate among patients with febrile neutropenia (FN), accounting for approximately 60% of blood cancer patients, is approximately 20%. Therefore, It is crucial to promptly and correctly assess and treat the situation, as any delay in starting the right antibiotic treatment could have fatal consequences [3].

Guidelines suggest treating these patients with broad-spectrum beta-lactam antibiotics such as piperacillin/tazobactam or cefepime [1,4,5]. However, the efficacy of beta-lactam antibiotics depends on the time during which the free plasma concentration of the antibiotics exceeds the minimal inhibitory concentration (MIC) [6]. Recent studies have suggested that standard dosage regimens of bolus infusion for beta-lactam antibiotics may not be ideal for achieving pharmacokinetic/pharmacodynamic (PK/PD) targets in patients [7]. For instance, beta-lactam antibiotics may experience changes in pharmacokinetic parameters during sepsis and septic shock, leading to insufficient concentrations of the drugs [8,9]. Unlike conventional intermittent infusion lasting no longer than 30 min, administration through prolonged (extended) intravenous infusion, either as an extended infusion (infusing the antibiotic over at least half of the dosing interval) or as a continuous infusion, results in a continuous and sustained beta-lactam concentration, which aligns with the pharmacodynamics of these drugs. Beta-lactam antibiotics administered via extended infusion have been shown to achieve PK/PD targets at a higher rate and improve the prognosis of patients with critical illness, thereby improving mortality, according to a meta-analysis [10]. The sepsis guidelines suggest using a prolonged (extended) infusion of beta-lactams for maintenance (after an initial bolus) over conventional bolus infusion for patients with sepsis or septic shock [11].

On the other hand, it has been shown that standard beta-lactam antibiotics may not achieve a sufficient time above the MIC in febrile neutropenic patients [12]. Patients with febrile neutropenia also differ in terms of increased volumes of distribution and drug clearance [13]. For instance, PIPC/TAZ exhibited altered pharmacokinetics for piperacillin in febrile neutropenic patients due to observed high volumes of distribution and clearance. Standard intermittent dosing of 4.5 g of piperacillin/tazobactam (via intravenous bolus infusion every 8 h) resulted in suboptimal antibiotic exposure and was therefore not sufficient [13,14]. Recent studies have suggested that extended infusion of beta-lactam antibiotics may be more effective than bolus infusion for treating febrile neutropenia [15,16,17,18,19]. Extended infusion involves administering an antibiotic over a longer period of time, typically 3–4 h, as opposed to a short bolus infusion over 30 min or less. This allows for more sustained and consistent levels of the antibiotic in the bloodstream, which may lead to better treatment outcomes. In addition, for bacteria with a high MIC, extended administration is recommended as the time above the MIC cannot be maintained [20], which is an important concept in the treatment of resistant strains.

To further investigate the potential benefits of extended or continuous infusion, a systematic review of clinical outcomes was conducted, comparing extended or continuous infusion of beta-lactam antibiotics with bolus infusion in patients with high-risk febrile neutropenia. The review aimed to provide important insights into the optimal dosing regimen for beta-lactam antibiotics in this patient population and to help improve treatment outcomes and reduce mortality rates.

## 2. Methods

### 2.1. Sources and Searches for Studies

An investigator developed a search strategy. Three databases, namely, PubMed, EMBASE, and Ichushi, were searched until 26 March 2023. We searched the terms “clinical study”, “epidemiologic studies”, “name of antibiotics”, “beta-lactams”, “neutropenia”, “drug administration schedule”, “administration and dosage”, “prolonged”, “extended”, “neoplasms”, “bone marrow transplantation”, “cancer”, “tumor”, “malignancy”, “carcinoma”, “leukemia”, “randomized controlled trial”, “controlled clinical trial”, placebo”, “clinical trials as topic”, “randomly”, NOT “animals”, and “humans”. The Appendix A section of this report includes specific and detailed information regarding the search strategies employed in three separate databases for this systematic review.

This systematic review was conducted according to the Preferred Reporting Items for Systematic reviews and Meta-Analyses (PRISMA) [21] and Meta-analysis Of Observational Studies in Epidemiology (MOOSE) guidelines [22]. The review protocol was recorded on 27 May 2022, on PROSPERO with the CRD number 42022333119.

### 2.2. Study Selection

As part of our systematic review, we took care to include randomized controlled trials (RCTs) and observational studies in any language that reported clinical failure, all-cause mortality, or adverse events in extended or continuous infusions of beta-lactam antibiotics in patients with hematological febrile neutropenia. In order to maintain the focus of our investigation, we excluded studies that assessed an un-estimated number of critically ill patients with neutropenia, such as those in the intensive care unit. To ensure consistency and accuracy, two investigators independently assessed the full texts of the articles, and any discrepancies were addressed and resolved with the assistance of a third investigator. It is important to note that the controls in these studies were administered bolus infusions of beta-lactam antibiotics.

### 2.3. Outcomes of Our Study

The main focus of this systematic review was to investigate and report on the primary outcomes of clinical failure, all-cause mortality, and adverse events. We have defined “clinical failure” as “not achieving clinical response” in our study. It is important to note that the definition of clinical response varied across the studies analyzed in this review. To ensure accuracy and consistency, the clinical responses were carefully extracted by the authors and are explained in greater detail in the “Characteristics of Studies” in Section 3.

### 2.4. Data Extraction from Each Study

As part of our systematic review, we took care to ensure the accuracy and completeness of our data by having two investigators independently extract the following information from each study: country of origin, published year, sample size, and type of beta-lactam antibiotic. Any discrepancies that arose during the extraction process were carefully reviewed and resolved through discussion with an additional investigator, further enhancing the reliability and validity of our results.

### 2.5. Risk of Bias Assessment for Each Study

To ensure the validity and reliability of our results, we employed a rigorous process for assessing the risk of bias in the studies included in this systematic review. Two investigators independently assessed the risk of bias, using the Cochrane risk-of-bias tool (RoB) for randomized trials and the Risk of Bias Assessment tool for Nonrandomized Studies (RoBANS) for the controlled observational studies [23]. Any discrepancies or disagreements that arose during the assessment process were resolved through discussion with a third investigator.

With RoB, we evaluated seven domains of bias, including random sequence generation, allocation concealment, blinding of participants and personnel, blinding of outcome assessments, incomplete outcome data, reporting selection, and other potential sources of bias. The overall risk of bias for each of the seven domains was categorized as low, unclear, or high. Similarly, RoBANS consists of six domains that we assessed, including the selection of participants, confounding variables, measurement of exposure, blinding of outcome assessments, incomplete outcome data, and selective outcome reporting.

### 2.6. Statistical Analyses

We analyzed dichotomous data by calculating the risk ratio (RR) and the odds ratio (OR) for RCTs and retrospective studies, with the uncertainty in each result presented as 95% confidence intervals (CIs). A fixed- or random-effects model was used unless significant heterogeneity was observed (*p* < 0.1 or I^2^ > 50%), where the random-effects model was used. We analyzed the data using Review Manager 5.4, released by Cochrane.

### 2.7. Certainty of Evidence

To ensure the quality of our findings, we evaluated the certainty of evidence of this systematic review using the GRADEpro GDT (guideline development tool) software (https://www.gradepro.org/, accessed on 1 January 2023) [24]. The GRADE framework considers various factors, including the study design, risk of bias, directness of outcomes, heterogeneity, precision within results, bias due to publication, estimate effect, dose-response relationship, and confounders when assessing the certainty of evidence. By taking all of these factors into account, we were able to determine the overall GRADE of the evidence obtained from our review. This GRADE can range from high to very low certainty of evidence, providing readers with an objective and comprehensive understanding of the strength and reliability of our findings.

## 3. Results

We performed a comprehensive search and retrieved 3391 citations, out of which 3376 records were excluded as they were not related to the study objective or were duplicates of other databases. After reviewing the full texts of fifteen articles, five studies [15,16,17,18,19] were selected for inclusion (Figure 1). We excluded six single-arm studies, one case report, one pharmacokinetics/pharmacodynamics study, and two studies that did not provide an estimated number of febrile neutropenia (FN) patients. The final selection included five articles published between 2014 and 2022. In total, 701 patients were included in this review.

### 3.1. The Results of Characteristics of Studies

Two randomized controlled trials (RCTs) were conducted in Israel and the USA [15,17]. One RCT was conducted in Mexico [19]. Two retrospective studies were conducted in Spain and the USA [16,18]. The sample sizes ranged from 63 [17] to 193 [18] (Table 1).

As stated in Section 2 we have defined “clinical failure” as “not achieving clinical response”. We defined clinical response as treatment success [16], overall response [15], clinical response [17], and defervescence by 24 h [18]. Treatment success after 5 days of treatment [16] was defined as follows: (i) a drop in body temperature to 37.5 °C leading to a ≥24 h fever-less state; (ii) resolution or improvement in the clinical signs and symptoms of infection when there had been any; (iii) the absence of persistent or breakthrough bacteremia; and (iv) no additional antibiotic treatment introduced because of an unsatisfactory clinical evolution. In another study, overall response [15] on day 4 post-symptom onset was defined as a composite of four criteria: (i) resolution of fever for at least 24 h; (ii) microbiological eradication (for microbiologically documented infection) and sterile cultures on days 3 and 4; (iii) clinical response (for clinically documented infection) and resolution of signs and symptoms of infection; and (iv) no need for a change in the antibiotic regimen (the addition of an aminoglycoside or a fluoroquinolone within 48 h of initiating treatment was not considered treatment failure). Clinical response in the other studies [17,19] was defined as follows: in one study [17], it was defined as defervescence within 72 h, while in another study [19], it was defined as an improvement in the signs and symptoms of infection at 72 h.

### 3.2. The Result of Risk of Bias Assessment, GRADE, and Meta-Analyses

The risk of bias assessment data for three RCT and two retrospective studies are graphically presented in Figure 2.

For extended infusion and continuous infusion, the RR (95% CI) for clinical failure was 0.74 (0.53, 1.05) and heterogeneity (χ^2^ = 1.98; *p* = 0.37; I^2^ = 0%) was observed in 3 RCTs (Figure 3a). For extended infusion, the RR (95% CI) for clinical failure was 0.73 (0.47, 1.14) and heterogeneity (χ^2^ = 1.97; *p* = 0.16; I^2^ = 49%) was observed in 2 RCTs (Figure 3b). For continuous infusion, the RR (95%CI) for clinical failure was 0.76 (0.44, 1.31) in 1 RCT (Figure 3c). Furthermore, in 2 retrospective studies evaluating extended infusion for clinical failure, the OR (95% CI) was 0.14 (0.02, 1.17), and heterogeneity (χ^2^ = 4.23; *p* = 0.04; I^2^ = 76%) was observed (Figure 3d).

In our systematic results, although the RR or the OR for clinical failure did not reach statistical significance, we observed a tendency toward favorable outcomes for clinical failure with an extended infusion or continuous infusion.

For extended infusion and continuous infusion, the RR (95% CI) for all-cause mortality was 1.25 (0.44, 3.54) and heterogeneity (χ^2^ = 0.78; *p* = 0.68; I^2^ = 0%) was observed in 3 RCTs (Figure 4a). For extended infusion, the RR (95%CI) for all-cause mortality was 1.36 (0.44, 4.25) and heterogeneity (χ^2^ = 0.62; *p* = 0.43; I^2^ = 0%) was observed in 2 RCTs (Figure 4b). For continuous infusion, the RR (95% CI) for all-cause mortality was 0.76 (0.05, 11.96) in 1 RCT (Figure 4c). Furthermore, in 2 retrospective studies evaluating extended infusion for all-cause mortality, the OR (95% CI) was 1.00 (0.44, 2.23) and heterogeneity (χ^2^ = 1.47; *p* = 0.22; I^2^ = 32%) was observed (Figure 4d). Only 1 RCT evaluated adverse events, with an RR (95% CI) of 0.46 (0.13, 1.65) (Figure 5). In our systematic review, when considering all-cause mortality and adverse events, the RR or the OR crossed 1 with a 95% CI. This indicates that extended infusion or continuous infusion was not found to be inferior to bolus infusion in terms of these outcomes.

According to the GRADE analysis for clinical failure in the 2 or 3 RCTs, the certainty of the evidence was “low” because the 95% CI reached beyond 1.0 owing to the small sample size. For all-cause mortality in 2 or 3 RCTs, the certainty of the evidence was also “low” because of crossing the non-significant line and the wide 95% CI with a small sample size. In the retrospective studies, clinical failure and all-cause mortality had a “very low” certainty of evidence owing to the study design (Table 2).

## 4. Discussion

In this systematic review, we examined extended infusion, continuous infusion, and bolus infusion of beta-lactam antibiotics in patients with hematological cancer and febrile neutropenia. We identified 2 RCTs and 2 observational studies (comprising 701 patients with distinct febrile neutropenia) comparing bolus infusion with extended infusion or continuous infusion of beta-lactam antibiotics. The trials had variable designs and criteria for clinical response in both study arms. In our systematic review, we found that extended infusion or continuous infusion did not reduce all-cause mortality rates or increase adverse events compared with bolus infusion of beta-lactam antibiotics. However, extended infusion and continuous infusion showed a positive impact on clinical failure. The study on patients with hematological cancer showed a higher mortality rate in these patients than those without cancer, and extended infusion may not have made a difference. For example, in the systematic review of extended infusion for sepsis [10], a total of 2196 articles were identified and reviewed. From those, 22 studies were selected for the meta-analysis, which included a total of 1876 patients. The findings revealed that prolonged (extended) infusion was linked to a lower risk of all-cause mortality in comparison with short-term infusion. There may be various reasons for the different results, such as the difference in the number of studies conducted, as well as the fact that the study was conducted on cancer patients, which could have affected the outcome. In the future, there is a possibility that these results may change by analyzing in more detail whether the all-cause mortality includes infection-related deaths or deaths of cancer patients. On the other hand, the development of febrile neutropenia during the course of chemotherapy is not only a life-threatening complication but can also lead to a decision to reduce the chemotherapy intensity in subsequent treatment cycles [25]. It is associated with significant morbidity and mortality and can lead to a decision to reduce or delay subsequent chemotherapy doses, which can have implications for treatment efficacy [26]. Thus, better clinical responses and fewer adverse events with the help of extended infusion are noted as good outcomes for patients undergoing chemotherapy.

Extended infusion has a good PK/PD outcome in critically ill patients with septic shock [27] or febrile neutropenia [28,29,30] and also in cases involving carbapenem-resistant (CR) *Enterobacteriaceae* (CRE) or CR *Pseudomonas aeruginosa* [31]. Recently, increasing numbers of patients with hematologic malignancies have been developing CRE infections [32], and extended infusion should be one of the treatment options. The period of time within the dosing interval of beta-lactam antibiotics during which free drug concentrations surpass the MIC is closely linked to the eradication of the targeted organisms. Probability of target attainment (PTA) analysis assesses the extent of plasma exposure achieved with an antibiotic dosing regimen in a population of patients, comparing it to the desired exposure needed for effectiveness relative to the MIC of a particular pathogen. In Gram-negative rods with high MICs, the standard dose of beta-lactam does not reach the PTA. Pharmacodynamic models that predict clinical responses based on specific pharmacological targets can help to identify the optimal antibiotic dosing strategy. These models take into account various factors, such as dosage, interval, and infusion time, to estimate the probability of achieving the desired time above the MIC. Furthermore, these models can be customized to specific patient groups or local resistance patterns by considering the distribution of the pathogen–antibiotic MIC. Multiple studies have shown that using extended infusion methods instead of traditional intermittent dosing is more effective for treating infections caused by *Pseudomonas aeruginosa* and other Gram-negative bacteria with higher MICs [33,34,35]. Therefore, in the future, we believe there is a high likelihood of the emergence of resistant bacteria in febrile neutropenic patients, and extended infusion may be helpful.

In addition to an extended infusion of beta-lactam, a 24-h continuous administration of beta-lactam has also been advocated. Randomized controlled studies and several systematic reviews have been conducted on sepsis [36,37,38], but the conclusions are inconclusive as there are cases where there is no difference in mortality, and there are cases where there is a difference in cure rates. There is limited research on continuous administration in patients with febrile neutropenia [19], and further data accumulation is needed.

Another option for empirical antibiotic therapy is combination therapy involving extended infusion of beta-lactams with aminoglycosides (AGs). Theoretically, as resistance in Gram-negative bacteria continues to increase, the use of AGs with beta-lactam antibiotics should also be considered. Two studies in Greece and Italy [39,40,41] have reported a better mortality rate with combination therapy using beta-lactam with AG or colistin for carbapenems-producing *Klebsiella pneumoniae*. Additionally, a study in Spain (the AMINOLACTAM study) reported an improvement in the mortality rate in patients with bloodstream infections when AG was added to the beta-lactam treatment [42]. This study included 27.7% multi-drug resistant (MDR) pathogens such as extended-spectrum beta-lactamase-producing Enterobacteriaceae and MDR *P. aeruginosa*. In such cases, an extended infusion of beta-lactams is necessary due to the high MIC. Furthermore, several studies have shown that combination therapy, such as the combination of beta-lactams with aminoglycosides, is associated with lower mortality rates in the treatment of resistant pathogens such as CRE [43]. The clinical impact of combination therapy in a systematic review of FN in 2013 was limited [44]. We are currently reviewing beta-lactam with AGs in a systematic review on PROSPERO with CRD number 42022379480. Although our systematic review did not provide information on antibiotic-resistant bacteria, it is more compelling to consider the utilization of extended infusion of beta-lactams with AG when there is a projected burden of antibiotic resistance in future clinical studies.

Regarding safety, only one RCT was found [15], but there was no difference between the two groups. According to a systematic review of sepsis [10], there were no significant differences in reported adverse events between the groups compared (7 RCTs; 980 patients; RR 0.88; 95% CI 0.71–1.09; I^2^ = 0%). Therefore, there is a possibility that extended infusion can be used safely, and although there is no difference in mortality rates, considering its favorable outcome, it may be worth considering its use.

There were certain limitations in this study. The trials included in this review were of different designs, ranging from randomized controlled trials (RCTs) to observational studies. Moreover, the definition of clinical response varied across studies. Nevertheless, the authors considered defervescence, in addition to microbiological eradication, as an important clinical outcome. This is because a previous study demonstrated that empirical antibiotic therapy can be safely discontinued in clinically stable neutropenic patients without waiting for the recovery of neutrophil counts in order to proceed with the next chemotherapy cycle [45].

The second limitation arises from the fact that varying doses of cefepime were administered in the bolus infusion arm and extended infusion arm [18]. This disparity in dosing regimens has the potential to influence the clinical outcomes.

While the extended infusion showed good efficacy against strains such as CRE and CR *P. aeruginosa*, our review could not mention resistant strains due to the lack of previous studies. However, a global report under the surveillance of the World Health Organization revealed that third-generation cephalosporin-resistant or carbapenem-resistant (CR) *Enterobacteriaceae* are common in the USA and Spain [46], indicating that resistant Gram-negative strains may also be included in these studies. Therefore, further research is required to evaluate the efficacy of extended infusions against resistant strains in cancer patients. Despite the low heterogeneity of the studies in this research and the potential for extended infusion of beta-lactam antibiotics to have a better outcome in the future, we believe that more RCTs are necessary to enhance the quality of systematic reviews. Currently, there are few RCTs and one observational study on extended infusion; however, another RCT (NCT04233996) in Spain is underway [47].

## 5. Conclusions

Although our review did not find any significant difference in mortality between extended infusion and bolus infusion of beta-lactam antibiotics, extended infusion may still be a preferable option due to its favorable outcomes, including a shorter duration of fever and fewer adverse events. Furthermore, although the evidence is limited, there is a possibility that extended infusion or continuous infusion can be used safely, similar to bolus infusion. While the evidence for these outcomes is limited by the small number of studies, the findings suggest that extended infusion has potential benefits for patients with hematological febrile neutropenia. Nonetheless, given the limitations of the available evidence, further randomized controlled trials are needed to provide a more definitive understanding of the effectiveness of extended infusion versus bolus infusion. Moreover, in the future, it is expected that it will be more effective, especially as infections caused by Gram-negative rods with high MICs become more frequent.

## Figures and Tables

**Figure 1 antibiotics-12-01024-f001:**
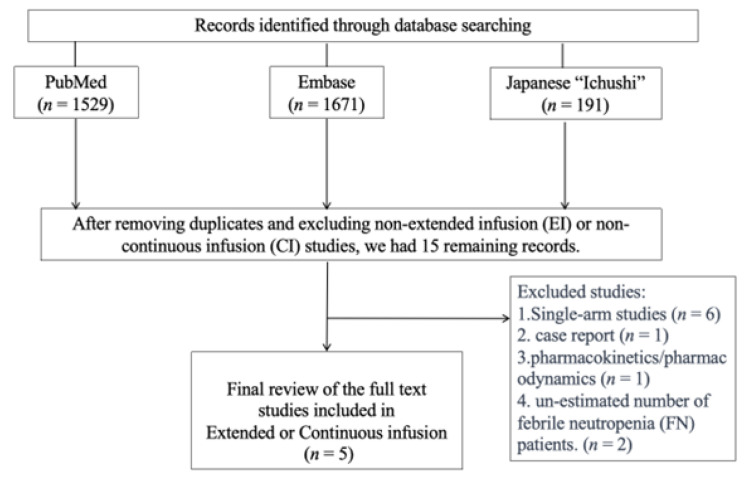
Identification process for eligible studies.

**Figure 2 antibiotics-12-01024-f002:**
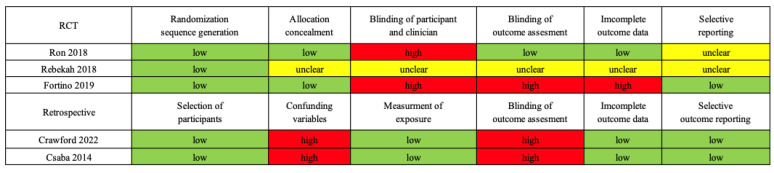
Summary of risk of bias in two randomized controlled trials and two retrospective studies. The risk of bias included randomization sequence, concealment, blinding of participants and clinicians, incomplete outcome data, selective reporting, and others in RCTs. The risk of bias included selection of participants, confounding variables, measurement of exposure, blinding of outcome assessment, incomplete outcome data, and selective outcome reporting in the retrospective studies. Green, low risk of bias; yellow, unclear risk of bias; red, high risk of bias. Abbreviation: RCT, randomized controlled study [15,16,17,18,19].

**Figure 3 antibiotics-12-01024-f003:**
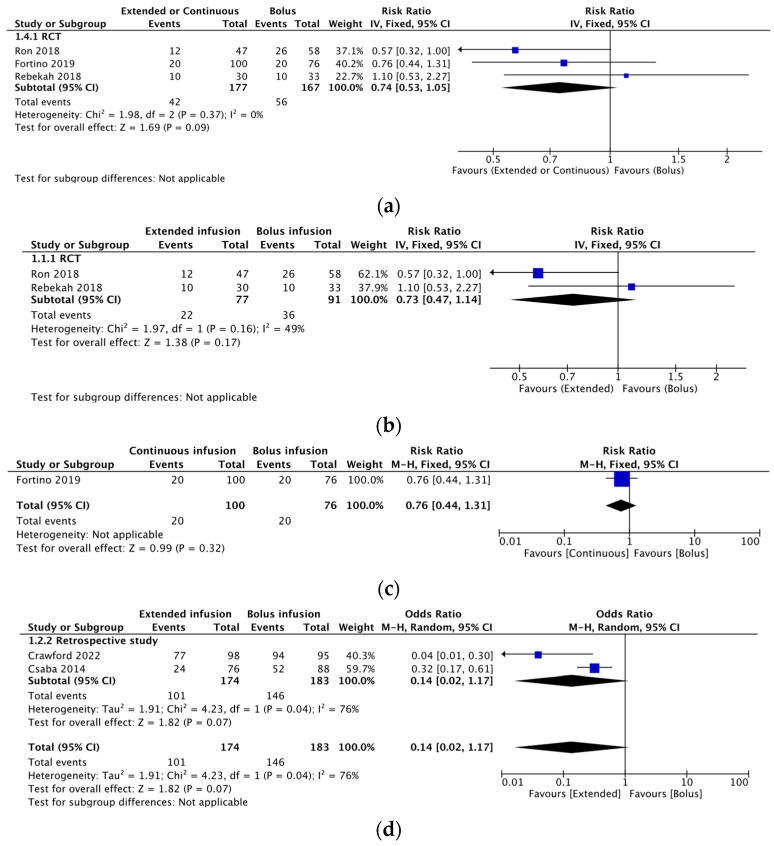
(**a**) Summary of clinical failure associated with extended and continuous infusion compared with that associated with bolus infusion in three RCTs. (**b**) Summary of clinical failure associated with extended infusion compared with that associated with bolus infusion in two RCTs. (**c**) Summary of clinical failure associated with continuous infusion compared with that associated with bolus infusion in one RCT. (**d**) Summary of clinical failure associated with extended infusion compared with that associated with bolus infusion in two retrospective studies. Abbreviations: CI, confidence interval; RCT, randomized controlled trial [15,16,17,18,19].

**Figure 4 antibiotics-12-01024-f004:**
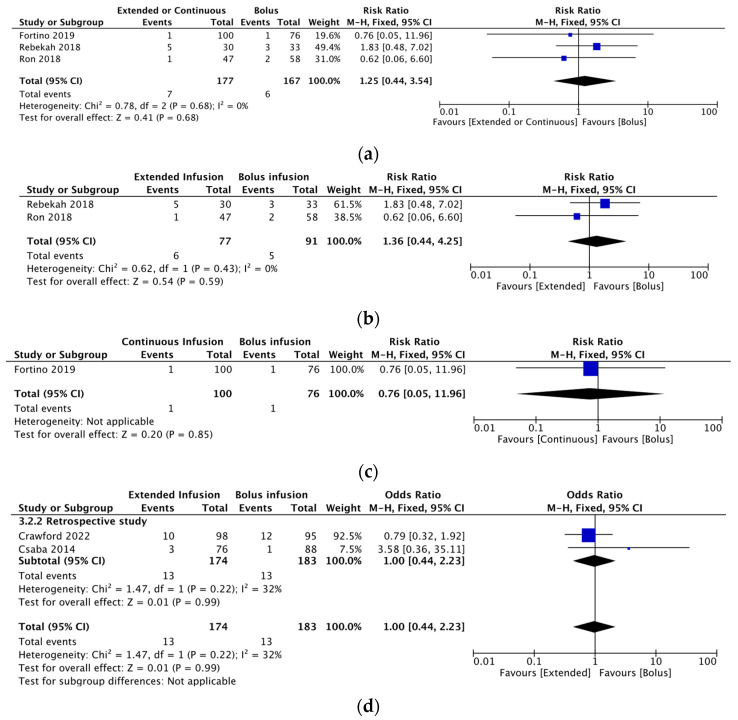
(**a**) Summary of all-cause mortality estimates associated with extended and continuous infusion compared with those associated with bolus infusion in three RCTs. (**b**) Summary of all-cause mortality rates associated with extended infusion compared with those associated with bolus infusion in two RCTs. (**c**) Summary of all-cause mortality rates associated with continuous infusion compared with those associated with bolus infusion in one RCT. (**d**) Summary of all-cause mortality rates associated with extended infusion compared with those associated with bolus infusion in two retrospective studies. Abbreviations: CI, confidence interval; RCT, randomized controlled trial [15,16,17,18,19].

**Figure 5 antibiotics-12-01024-f005:**
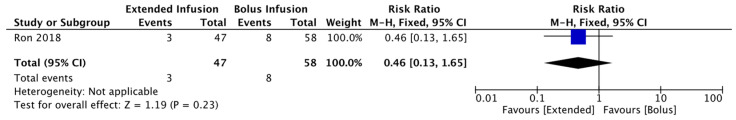
Summary of adverse event rates associated with extended infusion compared with those associated with bolus infusion in one RCT. Abbreviations: CI, confidence interval; RCT, randomized controlled trial [15].

**Table 1 antibiotics-12-01024-t001:** Characteristics of the included randomized controlled trials and retrospective studies.

Articles	Published Country	Study Design	Patient Characteristics/ Type of Beta-Lactam Antibiotics	Intervention	Comparison
Crawford 2022 [18]	USA	Retrospective/ single center	Hematologic malignancy (AML, 78.8%; ALL 21.2%)	EI (*n* = 98); CFPM 1 g/4 h q8h	BI (*n* = 95); CFPM 2 g/30 min q8h
Fortino 2019 [19]	Mexico	RCT/single center	Hematologic malignancy/children hematological malignancy: 20.5%; solid tumor: 79.5%	CI (*n* = 100); PIPC/TAZ 75 mg/kg bolus, followed by 300 mg/kg/day over 24 h	BI (*n* = 76); PIPC/TAZ 300 mg/kg/day divided into 4 doses/30 min
Ron 2018 [15]	Israel	RCT/single center	Hematologic malignancy (AML/MDS: 35.8%; lymphoma 27.6%; multiple myeloma: 34.1%)	EI ITT (*n* = 47); PP (*n* = 43); PIPC/TAZ 4.5 g/4 h q8h and CAZ 2 g/4 h q8h (if penicillin allergy)	BI ITT (*n* = 58); PP (*n* = 48); PIPC/TAZ 4.5 g/30 min q6h and CAZ 2 g/30 min q8h (if penicillin allergy)
Rebekah 2018 [17]	USA	RCT/single center	Hematologic malignancy (acute leukemia: 42.9%; lymphoma: 36.5%; multiple myeloma: 14.3%; and MDS: 4.8%)	EI (*n* = 30); CFPM 2 g/3 h q8h	BI (*n* = 33); CFPM 2 q/30 min q8h
Csaba 2014 [16]	Spain	Retrospective/ single center	Hematologic malignancy (acute leukemia: 40.9%; lymphoma: 28.0%)	EI (*n* = 76); MEPM 1 g/4 h q8h	BI (*n* = 88); MEPM 1 g/30 min q8h

Abbreviations: BI, bolus infusion; CAZ, ceftazidime; CFPM, cefepime; Crcl, creatinine clearance; CI, continuous infusion; EI, extended infusion; ITT, intention-to-treat; MEPM, meropenem; PIPC/TAZ, piperacillin/tazobactam; PP, per-protocol; CR, carbapenem-resistant; FN, febrile neutropenia; MIC, minimal inhibitory concentration; AML, acute myeloid leukemia; ALL, acute lymphoblastic leukemia; MDS, myelodysplastic syndrome.

**Table 2 antibiotics-12-01024-t002:** Summary of findings for main comparison.

Summary of Findings					
Clinical Failure of Extended or Continuous Infusion Compared with Bolus Infusion for Febrile Neutropenia					
Patient or Population: Febrile Neutropenia Intervention: Extended or Continuous Infusion Comparison: Bolus Infusion					
Outcomes	Anticipated Absolute Effects * (95% CI)		Relative Effect (95% CI)	№ of Participants (Studies)	Certainty of the Evidence (GRADE)
	Risk with Placebo	Risk with Treatment Response			
Clinical failure with 3 RCTs	335 per 1000	248 per 1000 (178 to 352)	RR 0.74 (0.53 to 1.05)	344 (3 RCTs)	⨁⨁◯◯ Low ^a,b^
Clinical failure with extended infusion in 2 RCTs	396 per 1000	289 per 1000 (186 to 451)	RR 0.73 (0.47 to 1.14)	168 (2 RCTs)	⨁⨁◯◯ Low ^a,b^
Clinical failure with extended infusion in 2 retrospective studies	798 per 1000	356 per 1000 (73 to 822)	OR 0.14 (0.02 to 1.17)	357 (2 observational studies)	⨁◯◯◯ Very low ^a,b,c^
All-cause mortality in 3 RCTs	36 per 1000	45 per 1000 (16 to 127)	RR 1.25 (0.44 to 3.54)	344 (3 RCTs)	⨁◯◯◯ Low ^a,b^
All-cause mortality with extended infusion in 2 RCTs	55 per 1000	75 per 1000 (24 to 234)	RR 1.36 (0.44 to 4.25)	168 (2 RCTs)	⨁◯◯◯ Low ^a,b^
All-cause mortality with extended infusion in 2 retrospective studies	71 per 1000	71 per 1000 (33 to 146)	OR 1.00 (0.44 to 2.23)	357 (2 observational studies)	⨁◯◯◯ Very low ^a,b,c^

* The risk in the intervention group (and its 95% confidence interval) is based on the assumed risk in the comparison group and the relative effect of the intervention (and its 95% CI). CI: confidence interval; OR: odds ratio; RR: risk ratio. GRADE working group grades of evidence are as follows. High certainty: we are very confident that the true effect lies close to the estimate of the effect. Moderate certainty: we are moderately confident in the effect estimate; the true effect is likely to be close to the estimate of the effect, but there is a possibility that it is substantially different. Low certainty: our confidence in the effect estimate is limited; the true effect may be substantially different from the estimate of the effect. Very low certainty: we have very little confidence in the effect estimate; the true effect is likely to be substantially different from the estimate of the effect. Explanations: ^a^ The 95% confidence interval still includes 1. ^b^ The sample size is small. ^c^ This is due to retrospective studies.

## Data Availability

The data can be requested from the corresponding author.

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
