# Peer review of "Effectiveness of Extended or Continuous vs. Bolus Infusion of Broad-Spectrum Beta-Lactam Antibiotics for Febrile Neutropenia: A Systematic Review and Meta-Analysis"

_antibiotics, 2023, doi:10.3390/antibiotics12061024_

Round 1

Reviewer 1 Report

This is a systematic review and meta-analysis comparing extended infusion (EI) with bolus infusion (BI) of beta-lactams in patients with high-risk febrile neutropenia. The topic is relevant as there are data showing improved survival in sepsis patients when treated with EI or continuous infusion as compared to BI. 

Data from a small number of studies (two randomized controlled trials and two retrospective investigations) are analyzed in this review and meta-analysis showing no apparent benefit of EI over BI. The manuscript is largely readable. However, the structure and the line of argumentation need to be improved in a revision process.

The authors found 3,391 citations, of which 3,376 were excluded (figure 1). You do not show why only 15 of the intitially found citations were further assessed. It seems to me that your search criteria were not ideal. You excluded nine studies with continuous infusion of antibiotics. Why did you do this? Your results could have been more meaningful by including studies with continuous infusion. The PK/PD principles of continuous and EI of beta-lactams are similar. Assessing both continuous and EI compared to BI in high-risk febrile neutropenia could have been relevant for the medical community.

The discussion overall seems lengthy. I believe it should be more concise with less focus on resistant bacteria and aminoglycoside combination therapy as these issues were not part of the study.

Specific comments:

Line 34-38: you cite one retrospective study with 9,018 patients with solid cancer to build up an argument for EI. However, your study only includes patients with high-risk neutropenia. The reader needs to understand that these are different conditions. You could explain why this reference is important or omit it.

Line 81-89 (also see above): why did you chose such broad search terms that yielded > 3.000 results? You could have you used a more specific search strategy.

Line 95-97: why did you omit patients who were treated with continuous infusion (see above). I believe the results could have been more meaningful if you had included these.

Line 158-159: I believe it should be common practice not to praise oneself in a scientific manuscript. This phrase should be omitted.

Line 161-176: first, you should describe the studies at least in some detail. What patients were included? Were these mono- or multicentre studies? The entire passage is difficult to read. I appreciate that outcomes in the studies that you reviewed were defined differently. I believe you should first describe the findings in the studies and then come to your conclusion how you define your outcome of clinical response.

Line 178-186: this passage is again difficult to read. You describe RR and OR but you do not mention what you assess (it could be something like "the RR for all-cause mortality in patients treated with EI was .... as compared to patients treated with BI"). By reading this passage one cannot know whether an elevated RR is beneficial or unfavourable.

Figure 1 (also see above): why were all but 15 studies excluded in the primary assessment? I believe the search criteria could have been more stringent.

Table 1: you should describe in the results section and later discuss that in reference 18 different dosages were used for BI and EI. Different dosages may be a reason for different outcomes.

Figure 3: in figure 3a the EI is on the right side, in figure 3b it is on the left side. It would be clearer to have them on one side.

Figure 3 and 4: the labeling of a and b is inconsistent: in a it is "extended", in b it is "experimental". You should use the same labeling.

Line 226-227: you state you examined EI and BI in patients with cancer and febrile neutropenia. I believe your inclusion criteria were "high-risk neutropenia in hematological malignancy". This should be made clear (see also above comment to line 161-176).

Line 232-233: I cannot find in table 2 and in figure 3 that EI improved clinical response. The results seem to be non significant. 

Line 233-235: what study on patients with cancer are you referring to? 

Line 255-257: infections by pathogens with high MIC may be treated better with EI but EI per se does not prevent infections by CRE. This passage should be more precise.

Dear Editor,

As I outlined in my comments to the authors I would appreciate a review and meta-analysis including both EI and continuous infusion compared to BI. However, this would be a very major revision. You may decide whether this should be requested.

Best regards

Author Response

 Thank you for your feedback. I made the revised letter. Please check the followings.

This is a systematic review and meta-analysis comparing extended infusion (EI) with bolus infusion (BI) of beta-lactams in patients with high-risk febrile neutropenia. The topic is relevant as there are data showing improved survival in sepsis patients when treated with EI or continuous infusion as compared to BI. 

Data from a small number of studies (two randomized controlled trials and two retrospective investigations) are analyzed in this review and meta-analysis showing no apparent benefit of EI over BI. The manuscript is largely readable. However, the structure and the line of argumentation need to be improved in a revision process.

The authors found 3,391 citations, of which 3,376 were excluded (figure 1). You do not show why only 15 of the intitially found citations were further assessed. It seems to me that your search criteria were not ideal. You excluded nine studies with continuous infusion of antibiotics. Why did you do this? Your results could have been more meaningful by including studies with continuous infusion. The PK/PD principles of continuous and EI of beta-lactams are similar. Assessing both continuous and EI compared to BI in high-risk febrile neutropenia could have been relevant for the medical community.

The discussion overall seems lengthy. I believe it should be more concise with less focus on resistant bacteria and aminoglycoside combination therapy as these issues were not part of the study

Respond> Thank you for your feedback. We have conducted a meta-analysis for continuous infusion as well and have added a figure to the study. We have also presented separate sub-analyses for extended infusion and continuous infusion in other figure. Regarding Figure 1, we have thoroughly revised it to align with the PRISMA guidelines and provide more detailed information. Although our study did not provide specific information on resistant strain, we anticipate that the increasing prevalence of CRE and other gram-negative resistant strains in the future will necessitate the use of combination therapy involving extended infusion and aminoglycosides.

Specific comments:

Line 34-38: you cite one retrospective study with 9,018 patients with solid cancer to build up an argument for EI. However, your study only includes patients with high-risk neutropenia. The reader needs to understand that these are different conditions. You could explain why this reference is important or omit it.

Respond>Thank you for your feedback. I have revised the statement according to your comment. In a retrospective study (Reference: DOI: 10.1097/JIM.0000000000000146), it has been established that the 30-day mortality rate among patients with febrile neutropenia (FN), accounting for approximately 60% of hematological cancer patients, is around 20%. This finding highlights the significantly high mortality associated with FN in the context of hematologic malignancies.

Line 81-89 (also see above): why did you chose such broad search terms that yielded > 3.000 results? You could have you used a more specific search strategy.

Respond> Thank you for providing the information. We are conducting a systematic search in our systematic review to minimize bias. We have included a healthcare librarian as a team member, and we are performing comprehensive searches across multiple databases. Our search methodology adheres to the Cochrane guidelines. https://training.cochrane.org/handbook/current/chapter-04.

Line 95-97: why did you omit patients who were treated with continuous infusion (see above). I believe the results could have been more meaningful if you had included these.

Respond>Thanks for the feedback. I also added a continuous infusion in our meta-analysis.

Line 158-159: I believe it should be common practice not to praise oneself in a scientific manuscript. This phrase should be omitted.

Respond>Thank you for your feedback. I remove this sentence.

Line 161-176: first, you should describe the studies at least in some detail. What patients were included? Were these mono- or multicentre studies? The entire passage is difficult to read. I appreciate that outcomes in the studies that you reviewed were defined differently. I believe you should first describe the findings in the studies and then come to your conclusion how you define your outcome of clinical response.

Respond>Thank you for providing the additional information. We have conducted a meta-analysis on the results of the studies reported in Table 1, which provides the baseline characteristics of the patients and the outcomes observed in each study. The table also indicates whether the studies were conducted at a single center or involved multiple centers. As described in the methods section, we have defined clinical failure, all-cause mortality, and adverse events. The specific definitions used in each study are provided in Table 1. The combined results of the studies are presented in Figure 3,4,5.

Line 178-186: this passage is again difficult to read. You describe RR and OR but you do not mention what you assess (it could be something like "the RR for all-cause mortality in patients treated with EI was .... as compared to patients treated with BI"). By reading this passage one cannot know whether an elevated RR is beneficial or unfavourable.

Response: Thank you for your feedback. I apologize for writing a very difficult-to-read passage. I have made revisions based on your suggestion to define and analyze Clinical failure rather than Clinical response, as it provides a clearer understanding of the results. When Clinical failure was defined, both the Randomized Controlled Trial (RCT) using risk ratio and the Retrospective study using Odds ratio showed that extended infusion and bolus infusion resulted in fewer clinical failures.

Figure 1 (also see above): why were all but 15 studies excluded in the primary assessment? I believe the search criteria could have been more stringent.

Thank you for your feedback. As I mentioned before, conducting a systematic search resulted in a large volume of findings. However, this was necessary to ensure comprehensive literature coverage. I have updated Figure 1 to provide clear explanations for the exclusion criteria. Please refer to it for more details.

Table 1: you should describe in the results section and later discuss that in reference 18 different dosages were used for BI and EI. Different dosages may be a reason for different outcomes.

Respond> Thank you for your feedback. As you mentioned, in the study by Crawford et al. (2022, PMID: 36106434), different doses of cefepime were administered in the bolus infusion arm and extended infusion arm: cefepime 2gram q8h/0.5 h or cefepime 1gram Q8h/4 h. This difference in dosing regimens may potentially impact the clinical outcomes. I have acknowledged this limitation in the evidence level and mentioned it in line 378-381.

Figure 3: in figure 3a the EI is on the right side, in figure 3b it is on the left side. It would be clearer to have them on one side.

Thank you for your feedback. I have aligned all the Extended Infusion (EI) references to the left side in all figures.

Figure 3 and 4: the labeling of a and b is inconsistent: in a it is "extended", in b it is "experimental". You should use the same labeling.

Respond>Thank you for your feedback. I changed the extended.

Line 226-227: you state you examined EI and BI in patients with cancer and febrile neutropenia. I believe your inclusion criteria were "high-risk neutropenia in hematological malignancy". This should be made clear (see also above comment to line 161-176).

Respond>Thank you for your feedback. I modified the followings in Line 296-298.

In this systematic review, we examined extended infusion, continuous infusion, and bolus infusion of beta-lactam antibiotics in patients with hematological cancer and febrile neutropenia.

Line 232-233: I cannot find in table 2 and in figure 3 that EI improved clinical response. The results seem to be non significant. 

Response>Thank you for your feedback. As I previously mentioned, defining and visualizing the results using clinical failure as the defined outcome has provided a clearer understanding.

Line 233-235: what study on patients with cancer are you referring to? 

Respond> In the Grade analysis, we analyzed all the studies cited for each outcome. In this Grade analysis, we analyzed RCTs and retrospective studies for Clinical failure and All-cause mortality, respectively. The cited studies for Clinical failure are represented by figures 3 a, b, and d, while for All-cause mortality, they correspond to figures 4 a, b, and d.

Line 255-257: infections by pathogens with high MIC may be treated better with EI but EI per se does not prevent infections by CRE. This passage should be more precise.

Respond> Thank you very much for your feedback. As you pointed out, EI is treatment, not prevention. I modified the following sentence in the Line 327.

“extended infusion should be one of the option of the treatment. The proportion of time within the dosing interval of beta-lactam antibiotics where free drug concentrations surpass the MIC is closely linked to the eradication of the targeted organisms. Probability of target attainment (PTA) analysis assesses the extent of plasma exposure achieved by an antibiotic dosing regimen in a population of patients, comparing it to the desired exposure needed for effectiveness relative to the MIC of a particular pathogen.”

Reviewer 2 Report

The evaluation may not have provided clear information due to factors such as the different spectra of the antibiotics used in the studies included in the study for the purpose of review, and the fact that the endpoints of the studies were different even though they seemed similar. Summarizing information such as patient populations, age, underlying diseases, disease stage, etc. in the studies included for review purposes, with a table, and even comparing the studies in this respect can provide a more reliable evaluation of the homogeneity or heterogeneity of the population.

Author Response

Thank you for your feedback. I made the revised letter follwings.

The evaluation may not have provided clear information due to factors such as the different spectra of the antibiotics used in the studies included in the study for the purpose of review, and the fact that the endpoints of the studies were different even though they seemed similar. Summarizing information such as patient populations, age, underlying diseases, disease stage, etc. in the studies included for review purposes, with a table, and even comparing the studies in this respect can provide a more reliable evaluation of the homogeneity or heterogeneity of the population.

Respond>Thank you for the feedback. As you mentioned, this systematic review is among the first to focus on extended infusion or continuous infusion in patients with febrile neutropenia, and there is still a limited number of randomized controlled trials (RCTs) available. However, with further research, we may gain more insights into the spectrum of antimicrobial agents and patient backgrounds. Despite the limited number of studies, our findings suggest that extended infusion is associated with fewer clinical failures and fewer adverse effects compared to bolus infusion. These results should be considered in future clinical practice, even with the scarcity of studies. Although there are only five studies included, we have summarized each study in Table 1.

Round 2

Reviewer 1 Report

Thank you very much for your revision. You have adequately addressed most of the issues I had raised. I still have some minor issues:

Line 183 - 197. I believe your description should still be more precise. Every statement needs to contain all relevant information. This passage is still not well readable. It could be something like "for extended infusion and continuous infusion, the RR (95%CI) for clinical failure was ..., respectively."

Line 303-305: I still cannot see from table 2 and figure 3 that clinical failure was significantly reduced with extended or continuous infusion.

Table 1: you state that table 1 indicates whether studies were mono- or multicentric. However, I could not find this information. This information could be added in the "study design" column. The column "patient characteristics / type of beta-lactam antibiotics" seems not clear enough (e.g. you write "concurrent infusion of gram-negative... and solid". This needs revision). I suggest you give the information on the intervention and comparison in the respective columns so you can entirely focus on patient characteristics in one column and leave the antibiotics to the other columns.

Small mistakes (e.g. missing prepositions).

Line 165-197 not well readable to me but this may be more related to style than to obvious mistakes.

Author Response

Thank you for your feedback. I have made the necessary revisions once again. I have also added interpretations of the results to the "Results" section. T

Please see provided attachment.
